# Association of Religious End Time Beliefs with Attitudes toward Climate Change and Biodiversity Loss

Benjamin S. Lowe * , Susan K. Jacobson, Glenn D. Israel  and Anna L. Peterson

Department of Wildlife Ecology and Conservation, University of Florida, Gainesville, FL 32611, USA
* Correspondence: benlowe@ufl.edu

**Abstract:** Mobilizing communities for environmental sustainability often involves engaging with religious values and beliefs, which can exert powerful influences on the attitudes, norms, and behaviors of the majority of people worldwide. Christianity is the largest world religion and, in some contexts, has also been among the most skeptical of climate and environmental concerns. A popular explanation for this skepticism focuses on eschatological views (i.e., end time beliefs) and posits that if the earth is going to be destroyed someday, there is little point in conserving it now. Empirical evidence is lacking, however, on the extent to which such beliefs actually influence environmental attitudes. We surveyed Christian undergraduate students in the US (N = 1520) and found that belief in the imminent return of Jesus Christ was not significantly associated with variables tested regarding biodiversity loss or climate change. Furthermore, a plurality responded that the earth will be renewed at the end (43%), not destroyed (24%), and beliefs about the fate of the earth were generally not related to attitudinal measures—except for a slim minority of respondents with strongest views that the earth will be destroyed—but were significantly associated with political ideology and literalist views of Scripture. These findings suggest that end time views may not be a major obstacle—at least among younger American Christians—to promoting socio-ecological sustainability.

**Keywords:** American evangelicals; biblical literalism; Christianity; eschatology; Generation Z; global warming; political ideology

## 1. Introduction

Although often overlooked in the past, there is growing recognition that religion is integral to environmental sustainability and has significant roles to play, whether for better or worse, in addressing the twin Anthropocene challenges of biodiversity loss and climate change [1–4]. As an integral aspect of culture, society, and identity, religion plays important roles in shaping values, beliefs, attitudes, norms, and behaviors, including with respect to how the majority of people around the world perceive and relate to each other and the environment [2,3,5]. In recent decades, a growing body of research has been published in this area (e.g., Daw et al. [6]; Qian et al. [7]), and in 2012, a resolution was adopted at the World Conservation Congress calling for dialogue and cooperation with faith communities around environmental conservation. Numerous other efforts also exist in this area, including the World Wildlife Fund's Sacred Earth initiative, the Yale Forum on Religion and Ecology, and faith-based organizations such as Plant With Purpose, GreenFaith, National Religious Partnership for the Environment (USA), and A Rocha, which is an international network of Christian conservation organizations.

Of the major world religions, Christianity has the largest number of adherents and is one of the most widespread and influential, both globally and in the US [8,9]. Over fifty years ago, Western Christianity was famously blamed for being at the heart of "the historical roots of our ecological crisis" [10]. Since then, a common explanation that has developed for this phenomenon is tied to religious beliefs about the "end times" [11–15]. As the explanation goes, Christians in some traditions (e.g., evangelical Protestants) tend to

hold that we are currently living in the "end times", according to a particular interpretation of passages in the Bible [14]. In this interpretation, the end times will be marked by a worsening state of the world that culminates in a final war between good and evil (i.e., Armageddon), the return of Jesus Christ, the evacuation (i.e., Rapture) of faithful Christians to another realm referred to as heaven, and the eventual destruction of the earth in judgment by fire. There are various nuances within this perspective, including around the specific order and timing of steps, but the widespread assumption is that these beliefs lead their Christian adherents—and evangelicals in particular—to be careless and hostile toward climate and environmental concerns. This is of particular concern in the US because evangelicals play powerful roles in the Republican Party and thus help shape American politics [14].

It is important to note that prominent environmental leaders who identify as Christians, including evangelicals, have pushed back on these assertions and called for a more nuanced understanding of this faith community [16,17]. Nonetheless, existing research has typically found American evangelicals to be the most skeptical of all major religious groups when it comes to climate change and other environmental concerns [18]. Why is this the case and, as has often been proposed through the years, is it associated with Christian end time views? Our study provides empirical data to address these questions with implications for the growing efforts to promote socio-ecological sustainability among faith communities around the world.

### 1.1. Christian Perspectives on the End Times

End time and apocalyptic beliefs are a common—and often polarizing—feature of many religious traditions [19–21]. Christian eschatology (i.e., the study of the end times) focuses on trying to interpret various Bible passages that address the world's future. Although there are diverse interpretations, this future can be very broadly categorized into two interconnected components: (1) the return of Jesus Christ and (2) the fate of people and the rest of the earth. It is important to highlight that Christians diverge considerably in their interpretation of these end times and what they entail, and even how worthwhile it is to dwell on them [18].

In general, all major traditions of Christianity (Protestant, Orthodox, and Catholic) hold that Jesus Christ will return one day to bring peace and justice to the world and that it could happen anytime [22]. Some of the earliest Christians seem to have expected Jesus to return in their lifetimes (e.g., Mark 13, 1 Thessalonians 4), and many present-day Christians do so as well [23]. According to the Pew Research Center [24], 41% of Americans believe that Christ will likely return by 2050, and that increases to a peak of 58% for white evangelicals. In a more recent study, they found that 47% of Christians believe we are living in the end times, though only 14% say they believe the earth will deteriorate until Jesus returns [11].

What will happen before, during, and after the return of Christ, along with what happens to various people and the earth in general, is debated. The Bible does not provide a single or obvious roadmap of these events [25]. The passages that address the end times tend to be challenging to interpret and often written in highly allegorical styles (a literary genre also known as "apocalyptic literature" [19]). Generally, however, there are currently two major schools of thought among American Christians about what will happen to the earth in the end time: Dispensationalism (a form of Premillennialism) and Amillennialism. A third theory, Postmillennialism, used to be prominent among American Christians but has faded since the 20th Century World Wars and is not considered further here. A summary of these three distinct schools of thought is provided in File S2; for a more thorough history and typology of Jewish and Christian eschatology, see the work of Charles [26].

One key biblical passage that shapes end time views describes the earth as being burned by fire (2 Peter 3). Some, especially more conservative evangelical and fundamentalist Protestants, interpret this to mean that the earth will literally be burned up and destroyed. This view is commonly aligned with Dispensationalism. Dispensationalism was particularly widespread among American evangelicals in the 20th century and was popularized in part

by best-selling books such as *The Scofield Reference Bible* [27], *The Late Great Planet Earth* [28], and the *Left Behind* series [29], all of which have sold millions of copies [30].

Others point out, however, that the fire described in these passages is properly understood in context as a refining fire that will cleanse and purify the earth (as metal ore is purified in a furnace). This lines up with the direct comparison the passage makes with the story of a great flood (i.e., Noah's Ark). Although the story of Noah's Ark has an undeniably destructive element, its stated purpose is to cleanse the earth of evil while saving Noah's family and representatives of every species (see Genesis 6–9). In this latter view of the earth's fate, heaven is not a location somewhere else where God's people go after they die, but rather the biblical vision is of heaven "descending" to be united with a renewed earth [16,31]. For many Christian (including evangelical) theologians, heaven is a way of referring to where God lives; at the end time, God's space and our space will be reunited in peace (Revelation 21–22). This position is commonly aligned with the eschatological view known as Amillennialism (File S2), is currently taught by leading evangelical theologians [31], and is gaining popularity in many parts of Christianity today, especially among younger generations [32].

*1.2. The Relationship between End Time Beliefs and Environmental Conservation*

Eschatology has been described as the "most ecologically decisive component of a theological system" [33] (p. 159). In the context of Christianity, this can take various forms depending on the particular perspective. Amillennialism typically holds that the earth will be redeemed and that Christians are to work toward restoration in the meantime. This view has particular potential to support environmental concern [34] (Curry-Roper, 1990). In interviews with Christian environmentalists (i.e., members of the "creation care movement"), Bloomfield [22] found not only that respondents were no more fatalistic than the general American public, but also that their end time beliefs in the eventual renewal of the world provide them with hope and inspiration for their environmental efforts in the present.

In contrast, Dispensationalism could potentially lead adherents to expect and even welcome bad things happening in the world, as it could herald the return of Christ [13,34]. This view also provides little incentive to care for the earth given that it will be destroyed one day anyway [32,33]. A 2016 survey found that 14% of Americans, but 24% of evangelicals, believe that global warming is a sign of the end times, while 11% of Americans, and 26% of evangelicals, say that "the end times are coming, therefore we don't need to worry about global warming" [23]. This reflects what some have termed the "end time apathy hypothesis", which is a standard explanation given for why conservative Christians—and American evangelicals in particular—tend to be less concerned or supportive of environmental issues [12,18].

While numerous studies cite end time views as a potential obstacle among some Christians—and evangelicals in particular—to caring about longer-term environmental problems [35,36], few studies have analyzed them specifically [18]. One early study found measures of conservative eschatology to be the strongest religious predictors of overall disposition toward environmental concern among clergy and religious activists [37]. Another study found that belief in the return of Jesus Christ reduced support for government action to curb global warming by 10–20% [13]. A caveat about this latter study is that belief in the return of Christ is a core doctrine shared across major branches of Christianity and does not distinguish between Christian end time perspectives, though this belief may feature more prominently for some groups. A 2022 US survey found a modest bivariate relationship between end time beliefs and concerns about climate change, with those who believe we are currently in the end times less likely than others (51% versus 62%) to say that climate change is a very or extremely serious problem [11].

Recent qualitative studies have resulted in mixed and more nuanced findings on the role of end time beliefs in climate and environmental attitudes. One study found that eschatological views were often linked to the belief that God is sovereign and thus humans do not have the

power to destroy the earth [38]. Focusing on climate action—even if not inherently a bad thing to do—can be seen as a distraction from more urgent priorities, such as evangelization [38]. Additionally, some respondents were strongly opposed to international climate regulations out of concern about the development of a one-world government ruled by the Antichrist (a common concern in Dispensationalism that was popularized by Lindsey [28] and others). This aligns with Chaudoin et al. [39], who found evangelicals to be no different than other Americans in supporting domestic climate policies, but significantly less supportive of international climate policies and cooperation.

A major study on climate skepticism among American evangelicals differentiates between "hot" and "cool" Millennialists [12]. Hot Millennialists made up a minority of respondents who believe that the end of the world is near and that climate change may be evidence of this belief. Cool Millennialists, however, form the majority of respondents and think it is impossible to know when the end will come (and also do not think much about end time beliefs and controversies). Another major study on American evangelicals and climate change found that eschatology was not identified as a reason for opposing environmental concern [40]. Instead, obstacles to pro-climate and environmental attitudes focused on political factors, a general culture of scientific skepticism (stemming from creation–evolution debates), and the prioritization of competing concerns such as evangelization and abortion [40].

Based on this mixed context, our study provides empirical evidence clarifying whether and how specific end time beliefs may be related to attitudes about critical sustainability challenges of our time. We do this by conducting a quantitative survey of young US Christians to investigate two related research questions:

(1)  Are Christian end time beliefs associated with attitudes toward biodiversity loss?
(2)  Are Christian end time beliefs associated with attitudes toward climate change?

## 2. Methods

We surveyed undergraduate students (age 18–23; part of Generation Z as defined by Dimock [41]) at 35 Protestant Christian undergraduate institutions (colleges and universities) across the United States in February–April 2020. Most of these institutions are members of the Council for Christian Colleges and Universities (CCCU) and are broadly evangelical-aligned. CCCU institutions require their leadership and faculty to be professing Christians. Although not all of them require a similar faith commitment from their students, most students also identify as Christian.

The surveys were conducted through the online survey platform Qualtrics and disseminated by professors to their students in general education and introductory-level science courses. The classes used for this survey comprised students from a wide range of academic majors who are typically early in their undergraduate experience. Surveys were conducted early in the semester before any related topics were addressed in class.

A total of 1669 responses were collected from 35 campuses in 19 states (File S1) as part of a broader study on Christian attitudes about climate change (see the work of Lowe et al. [42] for additional results). The full surveys took approximately fifteen minutes to complete, and the overall response rate was 58%. This is a purposive sample, which enabled us to collect a large enough sample of Christians to make robust comparisons between numerous variables and subsets. It offers particular insight into the beliefs, attitudes, and opinions of younger Christians, which is an important demographic for understanding the future of American society and support for environmental sustainability. At the same time, this is not a probability-based sample, and caution is needed before drawing wider conclusions without further replication.

### 2.1. Instrumentation and Variables

The questions and response categories used in this study are drawn from established sources including the joint Yale–George Mason University Climate Change in the American Mind (CCAM) project, which has been surveying the US population biannually since

2008 [43]. In line with the CCAM project, our survey used the term "global warming" instead of "climate change", although we use both terms interchangeably for discussion purposes in this paper. While there is ongoing debate around which of these terms is less politically polarized, recent research suggests that they may currently be similar [44].

Specific survey items focused on how much of a priority it should be to address global warming were adapted from a study at similar religious institutions [45] and expanded to include biodiversity loss. Religion measures were used verbatim from both the CCAM project and a major study on religion and climate change conducted jointly by the American Academy of Religion and the Public Religion Research Institute [46]. Race and gender items were drawn from the CCAM project and US Census.

End time beliefs were measured using three questions. The first asks if respondents agree or disagree that Jesus Christ could return at any time. The second and third questions focus on two prominent and diverging eschatological views about the fate of the earth. The second question measures agreement with the proposition that the earth will be destroyed at the end, and the third asks if the earth will instead be renewed.

*2.2. Data Processing*

Data were analyzed using IBM's Statistical Program for the Social Sciences (SPSS Version 27). Responses of "don't know" were recoded as missing, and missing values analysis showed an arbitrary pattern with an overall missing rate of 5.79%. Missing values were estimated by multiple imputation procedures (ten imputations) using the MCMC algorithm with fully conditional specification [47,48]. Given that this study focuses on examining specific religious beliefs unique to Christianity, we only included those respondents who identified as Christian (approximately 91%) for subsequent analysis, which resulted in a final sample size of N = 1520.

A religiosity index was created that combines four variables: frequency of service attendance, how personally important religion is, whether the Bible is the highest authority for belief, and how personally important evangelism is to respondents. This scale is scored from a low of 4 to a high of 16 and was tested using Cronbach's alpha ($\alpha > 0.8$) and common factor analysis using principal axis factoring (PAF) [49]. Race was based on whether respondents identified as Hispanic, Black (non-Hispanic), White (non-Hispanic), or other (American Indian, Alaska Native, Asian Native Hawaiian or Pacific Islander, etc.).

The two end time belief items about the fate of the earth (destroyed or renewed) were combined into a single variable for analysis to enable direct comparisons to be made while separating out the small number of respondents who chose inconsistent responses, such as agreeing both that the earth will be renewed and destroyed. The forced-choice nature of these items does not include a neutral/do not know response category on end time belief questions, so respondents who truly lack settled views do not have an obvious option to select. This helps explain the minority of respondents who selected that they either disagreed or agreed with both options presented, which should be understood as mutually exclusive. To address this issue and ensure more consistent analysis, we combined the two separate items into a single "fate of the earth" variable that includes those who say they strongly or somewhat agree with one option, while strongly or somewhat disagreeing with the other. Respondents who selected that they agreed at some level with both (27%) and those who said they disagreed at some level with both (6%) were combined into a new "other" category. This third category conceptually includes those who do not hold a clear view on the fate of the earth.

At the same time, we recognize that end time beliefs can be complex and confusing, and their influence on attitudes may be greater for those with stronger or clearer views. Thus, we also created and tested an additional independent variable comprising only respondents who *strongly* agree that the earth will either be renewed or destroyed, while disagreeing with the alternative. This resulted in a subsample (38% of the full sample) comprising only those who express strong and consistent views on the fate of the earth.

*2.3. Analytical Approach*

To begin with, we tested bivariate relationships between the independent (biodiversity loss and climate change measures) and dependent variables using crosstabs with chi-square tests. This was to allow for more direct comparisons with existing studies that also tested bivariate relationships. SPSS does not offer pooled results across multiple imputations for chi-square tests, so pooling was accomplished in R using the "micombine.chisquare" function in the "miceadds" package, which pools chi-square results into an F-statistic [50–52]. Next, we used multinomial logistic regression models to test our research questions, starting with all potential predictors in the model and dropping the most non-significant variable before re-running the logistic regression. To check for suppressed relationships, we tested models with all the independent variables entered together, with just religion-related variables, and with just non-religion-related variables (and reported the pooled results that SPSS offered here across all imputations). This process of variable selection and elimination was iterated until we identified the best model (i.e., most parsimonious) with all significant variables using fit statistics including AIC/BIC scores. Since partisan identification and political ideology are distinct but highly correlated [53], we tested these variables in separate models and obtained consistently similar results. For the final models, we chose to focus on political ideology as it tended to return stronger results than partisan identity, which is not surprising given that our sample comprises younger respondents, who show signs of lower party affiliation in general [53].

## 3. Results

Of the 1520 Christian respondents, 70% identify as evangelical, which is expected given that the institutions we surveyed are generally aligned with the evangelical Protestant tradition (Table 1). A majority of respondents (70%) hold a nuanced theological position that the Bible is the word of God but that not all of it should be taken literally (i.e., it needs to be interpreted). In general, our sample population is highly religious, more politically conservative and more Republican than the American public [54], and majority white and female, all of which is unsurprising given the sample population [42].

**Table 1.** Distribution of variables tested in this study.

| Variable | Description | Pooled Distribution [a] |
|---|---|---|
| 2nd coming | The 2nd coming of Christ could take place at any time | Agree = 89%<br>Disagree = 11% |
| Fate of the earth (all) | Whether the earth is going to be destroyed or renewed at the end of time | Destroyed = 24%<br>Renewed = 43%<br>Other = 33% |
| Fate of the earth (strongly agree only) [b] | Whether the earth is going to be destroyed or renewed at the end of time (Subset of "fate of the earth" variable above with only respondents who strongly agree either that the earth is going to be destroyed or renewed at the end of time, while consistently disagreeing with the other view) | Destroyed = 39%<br><br>Renewed = 61% |
| Global warming happening | Do you think that global warming is happening? | Yes = 90%<br>No = 10% |
| Global warming cause | Assuming global warming is happening, do you think it is… | Caused mostly by humans = 76%<br>Not caused mostly by humans = 24% |
| Global warming worry | How worried are you about global warming? | Not at all worried = 14%<br>Not very worried = 26%<br>Somewhat worried = 42%<br>Very worried =18 % |
| Global warming priority—you | Do you think addressing global warming should be a low, medium, or high priority for you? | Low = 24%<br>Medium = 42%<br>High = 35% |
| Global warming priority—Christians | Do you think addressing global warming should be a low, medium, or high priority for Christians in general? | Low = 13%<br>Medium = 42%<br>High = 45% |

**Table 1.** *Cont.*

| Variable | Description | Pooled Distribution [a] |
|---|---|---|
| Global warming priority—government | Do you think addressing global warming should be a low, medium, or high priority for the government? | Low = 10%<br>Medium = 28%<br>High = 61% |
| Biodiversity loss worry | How worried are you about biodiversity loss/extinctions? | Not at all worried = 6%<br>Not very worried = 18%<br>Somewhat worried = 41%<br>Very worried = 35% |
| Biodiversity loss priority—you | Do you think addressing biodiversity loss/extinctions should be a low, medium, or high priority for you? | Low = 20%<br>Medium = 39%<br>High = 41% |
| Biodiversity loss priority—Christians | Do you think addressing biodiversity loss/extinctions should be a low, medium, or high priority for Christians in general? | Low = 9%<br>Medium = 41%<br>High = 50% |
| Biodiversity loss priority—government | Do you think addressing biodiversity loss/extinctions should be a low, medium, or high priority for the government? | Low = 8%<br>Medium = 29%<br>High = 64% |
| Evangelical identification | Would you describe yourself as "born-again" or evangelical? | Yes = 70%<br>No = 30% |
| View of bible | What is your view of the Bible?<br>(biblical literalism) | The word of God and to be taken literally, word for word = 21%<br>The word of God but not everything in the Bible should be taken literally, word for word = 70%<br>Written by humans and is not the word of God = 8% |
| Religiosity | Index variable combining service attendance, importance of religion, authority of the Bible, and importance of evangelism; Cronbach's $\alpha > 0.8$ | Mean = 13.3; SD = 2.9; range = 4–16 |
| Partisan identification | In general, do you think of yourself as…? | Republican = 50%<br>Democrat = 17%<br>Other/no party = 33% |
| Political ideology | In general, do you think of yourself as…? | Conservative = 50%<br>Centrist = 34%<br>Liberal = 16% |
| Race | Combined variable that includes respondents' self-identified racial category/categories and whether they also identify as Hispanic or Latino | White, non-Hispanic = 70%<br>Black, non-Hispanic = 6%<br>Hispanic = 9%<br>Other, non-Hispanic = 15% |
| Sex [c] | What is your sex? | Female = 61%<br>Male = 39% |

[a] N = 1520. [b] N = 580 for this subset variable. [c] Sex is presented as a binary variable based on the current approach used in CCAM surveys and the US Census; it does not account for all the sex or gender categories that respondents may identify with.

In terms of their views on climate change and biodiversity loss (Table 1), large majorities affirm that global warming is happening and primarily anthropogenic. There is greater variation when it comes to how worried respondents are about both global warming and biodiversity loss, as well as how much of a priority it should be to address these issues at different levels. Even so, strong majorities express at least some concern over both and say that addressing them should be at least a medium priority for themselves, Christians in general, and the government. Overall, respondents expressed higher levels of worry about biodiversity loss than global warming and also ranked biodiversity loss as a higher priority than global warming. Respondents ranked both issues as lower priorities for themselves to address but as progressively higher priorities for Christians in general followed by the government.

### 3.1. End Time Beliefs

Almost 90% of respondents reported the belief that Jesus Christ could return at any moment. Only 24% believe the earth will be destroyed by fire at the end, while a plurality (43%) aligns with the alternate view that the earth will be renewed at the end. The remaining third of our respondents expressed mixed/other views. Focusing only on those who *strongly*

agree with either of the two options (while consistently disagreeing with the alternative, thus eliminating the "other" category), the majority strongly agree that the earth will be renewed rather than destroyed.

3.1.1. Research Question 1: Are Christian End Time Beliefs Associated with Attitudes toward Biodiversity Loss?

Both bivariate and multivariate comparisons did not identify any significant relationships between beliefs in the imminent return of Christ or the fate of the earth (destroy versus renew) and any of the attitudinal measures around biodiversity loss (Files S3 and S4). Sociodemographic variables that were significant in bivariate comparisons include biblical literalism, partisan identification, political ideology, and race. Sex and evangelical identification were significantly associated with belief in the imminent return of Christ but not with views on the fate of the earth.

When compared using multinomial logistic regression models, race, sex, and evangelical identification were insignificant, and the best-fit models retained biblical literalism and political ideology. Those who hold that the Bible is the word of God but that not all of it should be taken literally are more likely to say that biodiversity loss should be a high priority and less likely to say it should be a medium or low priority, compared with those who say the Bible should be read literally (Tables 2 and 3). Political liberals and centrists are more likely to assert a high priority than conservatives (Tables 2 and 3).

When we focused more narrowly on the subset of our sample who express strong and consistent agreement with their view of the fate of the earth (N = 580 or 38% of the full sample), further analysis shows that end time beliefs are still not significantly associated with attitudes around biodiversity loss. On how much Christians should prioritize addressing biodiversity loss, the best model for this subset still includes biblical literalism and political ideology as the significant independent variables, with the latter having the strongest effect (File S5).

**Table 2.** Pooled logistic regression model results predicting how much of a priority it should be for Christians to address biodiversity loss (N = 1520).

| | Predictor | B | SE | F | df | Sig. | Exp (B) | Lower 95% CI | Upper 95% CI |
|---|---|---|---|---|---|---|---|---|---|
| Low priority | Bible = 1 (not God's word) | 0.383 | 0.413 | 0.861 | 1, 36.464 | 0.360 | 1.467 | 0.645 | 3.340 |
| | Bible = 2 (not all literal) | −0.491 | 0.228 | 4.650 | 1, 9546.683 | 0.031 | 0.612 | 0.392 | 0.956 |
| | Bible = 3 (literalist) | 0 | | | N/A (reference category) | | | | |
| | Political ideology = 1 (liberal) | −1.037 | 0.313 | 10.963 | 1, 331.989 | 0.001 | 0.355 | 0.192 | 0.656 |
| | Political ideology = 2 (centrist) | −1.076 | 0.236 | 20.780 | 1, 897.699 | 0.000 | 0.341 | 0.215 | 0.542 |
| | Political ideology = 3 (conservative) | 0 | | | N/A (reference category) | | | | |
| | Intercept | −0.954 | 0.197 | 23.393 | 1, 7278.143 | 0.000 | N/A | N/A | N/A |
| Medium priority | Bible = 1 | −0.294 | 0.311 | 0.897 | 1, 27.846 | 0.352 | 0.745 | 0.400 | 1.388 |
| | Bible = 2 | −0.284 | 0.139 | 4.182 | 1, 6919.576 | 0.041 | 0.753 | 0.574 | 0.988 |
| | Bible = 3 | 0 | | | N/A (reference category) | | | | |
| | Political ideology = 1 | −0.887 | 0.170 | 27.101 | 1, 869.640 | 0.000 | 0.412 | 0.295 | 0.575 |
| | Political ideology = 2 | −0.567 | 0.125 | 20.693 | 1, 1200.848 | 0.000 | 0.567 | 0.444 | 0.724 |
| | Political ideology = 3 | 0 | | | N/A (reference category) | | | | |
| | Intercept | 0.351 | 0.127 | 7.588 | 1, 15,556.227 | 0.006 | N/A | N/A | N/A |

Overall model: $\chi^2$ = 71.168–87.547; df = 8; *p* = 0.000; AIC = 104.031–113.801; BIC = 157.170–167.138. Reference category is "high priority"; N/A = Not Applicable

**Table 3.** Pooled logistic regression model results predicting how much of a priority it should be for the government to address biodiversity loss (N = 1520).

| | Predictor | B | SE | F | df | Sig. | Exp (B) | Lower 95% CI | Upper 95% CI |
|---|---|---|---|---|---|---|---|---|---|
| Low priority | Bible = 1 (not God's word) | 0.208 | 0.444 | 0.219 | 1, 55.943 | 0.642 | 1.231 | 0.511 | 2.962 |
| | Bible = 2 (not all literal) | −0.535 | 0.230 | 5.386 | 1, 3814.583 | 0.020 | 0.586 | 0.373 | 0.920 |
| | Bible = 3 (literalist) | 0 | | | N/A (reference category) | | | | |
| | Political ideology = 1 (liberal) | −1.534 | 0.401 | 14.662 | 1, 263.285 | 0.000 | 0.216 | 0.098 | 0.473 |
| | Political ideology = 2 (centrist) | −1.255 | 0.281 | 19.910 | 1, 84.739 | 0.000 | 0.285 | 0.164 | 0.496 |
| | Political ideology = 3 (conservative) | 0 | | | N/A (reference category) | | | | |
| | Intercept | −1.221 | 0.196 | 38.775 | 1, 1514.141 | 0.000 | N/A | N/A | N/A |
| Medium priority | Bible = 1 | 0.194 | 0.294 | 0.438 | 1, 38.769 | 0.512 | 1.215 | 0.678 | 2.177 |
| | Bible = 2 | −0.304 | 0.144 | 4.424 | 1, 8703.505 | 0.035 | 0.738 | 0.556 | 0.980 |
| | Bible = 3 | 0 | | | N/A (reference category) | | | | |
| | Political ideology = 1 | −1.119 | 0.197 | 32.321 | 1, 1748.494 | 0.000 | 0.327 | 0.222 | 0.480 |
| | Political ideology = 2 | −0.642 | 0.138 | 21.809 | 1, 324.901 | 0.000 | 0.526 | 0.402 | 0.689 |
| | Political ideology = 3 | 0 | | | N/A (reference category) | | | | |
| | Intercept | −0.228 | 0.130 | 3.087 | 1, 18,165.010 | 0.079 | N/A | N/A | N/A |

Overall model: $\chi^2$ = 90.019–105.403; df = 8; $p$ = 0.000; AIC = 102.218–118.275; BIC = 155.357–171.702. Reference category is "high priority"; N/A = Not Applicable

Consistent with previous results, those who say that the Bible is God's word but not all of it should be taken literally are approximately two-thirds less likely than biblical literalists to say that biodiversity loss should be a low versus high priority for Christians to address. Similarly, political centrists and liberals are less likely than conservatives to be in the low and medium priority groups. In response to the question of how much the government should prioritize addressing biodiversity loss, biblical literalism drops out, and political ideology remains as the only significant predictor (File S6). Compared with biodiversity loss models using the full sample, the directions of relationships in this subsample are consistent (with the exception of biblical literalism effect on priority for governmental action), but the effect sizes (as measured by exp(B)) tend to be larger.

### 3.1.2. Research Question 2: Are Christian End Time Beliefs Associated with Attitudes toward Climate Change?

While bivariate comparisons identified no significant relationships between belief about the imminent return of Christ and any of the global warming measures, views on the fate of the earth were associated with how much of a priority it should be for (1) Christians and (2) the government to address global warming. Biblical literalism, partisan identification, political ideology, and race were also significantly associated with climate attitudes.

When compared in multinomial logistic regression models, however, beliefs about the fate of the earth are no longer a significant predictor of how much of a priority global warming should be for Christians or the government. Race also loses its significance in the model, but biblical literalism and political ideology remain strongly associated with the prioritization of addressing global warming in both cases (Tables 4 and 5). Biblical literalists are about twice as likely to be in the low (versus high) priority category compared with those who say the Bible is God's word but not everything should be interpreted literally. Respondents who identify as politically liberal or centrist are more likely to be in the high priority category.

**Table 4.** Pooled logistic regression model results predicting how much of a priority it should be for Christians to address global warming (N = 1520).

|  | Predictor | B | SE | F | df | Sig. | Exp (B) | Lower 95% CI | Upper 95% CI |
|---|---|---|---|---|---|---|---|---|---|
| Low priority | Bible = 1 (not God's word) | 0.266 | 0.398 | 0.446 | 1, 28.764 | 0.510 | 1.305 | 0.588 | 2.893 |
|  | Bible = 2 (not all literal) | −0.682 | 0.196 | 12.117 | 1, 32,702.073 | 0.001 | 0.506 | 0.345 | 0.742 |
|  | Bible = 3 (literalist) | 0 | | | | N/A (reference category) | | | |
|  | Political ideology = 1 (liberal) | −1.925 | 0.315 | 37.384 | 1, 308.685 | 0.000 | 0.146 | 0.079 | 0.271 |
|  | Political ideology = 2 (centrist) | −1.303 | 0.204 | 40.679 | 1, 255.201 | 0.000 | 0.272 | 0.182 | 0.406 |
|  | Political ideology = 3 (conservative) | 0 | | | | N/A (reference category) | | | |
|  | Intercept | −0.61 | 0.173 | 0.124 | 1, 13,137.674 | 0.724 | N/A | N/A | N/A |
| Medium priority | Bible = 1 | −0.091 | 0.293 | 0.097 | 1, 57.910 | 0.757 | 0.913 | 0.512 | 1.628 |
|  | Bible = 2 | −0.268 | 0.148 | 3.289 | 1, 6049.454 | 0.070 | 0.765 | 0.572 | 1.022 |
|  | Bible = 3 | 0 | | | | N/A (reference category) | | | |
|  | Political ideology = 1 | −1.579 | 0.180 | 76.540 | 1, 1351.633 | 0.000 | 0.206 | 0.145 | 0.294 |
|  | Political ideology = 2 | −0.900 | 0.135 | 44.592 | 1, 209.762 | 0.000 | 0.406 | 0.312 | 0.530 |
|  | Political ideology = 3 | 0 | | | | N/A (reference category) | | | |
|  | Intercept | 0.714 | 0.140 | 25.123 | 1, 3093.556 | 0.000 | N/A | N/A | N/A |

Overall model: $\chi^2$ = 165.339–188.457; df = 8; *p* = 0.000; AIC = 103.486–121.970; BIC = 156.652–175.055. Reference category is "high priority"; N/A = Not Applicable

**Table 5.** Pooled logistic regression model results predicting how much of a priority it should be for the government to address global warming (N = 1520).

|  | Predictor | B | SE | F | df | Sig. | Exp (B) | Lower 95% CI | Upper 95% CI |
|---|---|---|---|---|---|---|---|---|---|
| Low priority | Bible = 1 (not God's word) | 0.121 | 0.485 | 0.062 | 1, 22.740 | 0.805 | 1.129 | 0.425 | 3.001 |
|  | Bible = 2 (not all literal) | −0.519 | 0.204 | 6.457 | 1, 27,730.832 | 0.011 | 0.595 | 0.399 | 0.888 |
|  | Bible = 3 (literalist) | 0 | | | | N/A (reference category) | | | |
|  | Political ideology = 1 (liberal) | −2.114 | 0.407 | 26.933 | 1, 159.405 | 0.000 | 0.121 | 0.054 | 0.269 |
|  | Political ideology = 2 (centrist) | −1.464 | 0.232 | 39.947 | 1, 457.693 | 0.000 | 0.231 | 0.147 | 0.364 |
|  | Political ideology = 3 (conservative) | 0 | | | | N/A (reference category) | | | |
|  | Intercept | −0.750 | 0.175 | 18.334 | 1, 7502.577 | 0.000 | N/A | N/A | N/A |
| Medium priority | Bible = 1 | 0.193 | 0.291 | 0.437 | 1, 57.157 | 0.511 | 1.212 | 0.681 | 2.157 |
|  | Bible = 2 | −0.294 | 0.147 | 3.992 | 1, 306,460.839 | 0.046 | 0.745 | 0.559 | 0.994 |
|  | Bible = 3 | 0 | | | | N/A (reference category) | | | |
|  | Political ideology = 1 | −1.889 | 0.233 | 65.714 | 1, 3135.066 | 0.000 | 0.151 | 0.096 | 0.239 |
|  | Political ideology = 2 | −0.884 | 0.151 | 34.044 | 1, 66.228 | 0.000 | 0.413 | 0.306 | 0.557 |
|  | Political ideology = 3 | 0 | | | | N/A (reference category) | | | |
|  | Intercept | −0.039 | 0.136 | 0.082 | 1, 1131.600 | 0.775 | N/A | N/A | N/A |

Overall model: $\chi^2$ = 172.357–198.653; df = 8; *p* = 0.000; AIC = 103.963–123.018; BIC = 157.102–176.103. Reference category is "high priority"; N/A = Not Applicable

When focused on the subset who express strong and consistent views about the fate of the earth (N = 580 or 38% of the full sample), further analysis does show significant associations between end time beliefs and views on how much of a priority addressing global warming should be for Christians and for the government (Table 6). Those who report strongly agreeing that the earth will be destroyed (15% of the full sample) instead of renewed (23% of the full sample) are more than twice as likely to say that addressing global warming should be a low versus high priority for Christians and for the government.

**Table 6.** Pooled logistic regression model results of those with strongly held beliefs only (N = 580), predicting how much of a priority it should be to address global warming.

| | **For Christians** | | | | | | | | |
|---|---|---|---|---|---|---|---|---|---|
| | **Predictor** | **B** | **SE** | **F** | **df** | **Sig.** | **Exp (B)** | **Lower 95% CI** | **Upper 95% CI** |
| Low priority | Fate = 1 (destroyed) | 0.775 | 0.290 | 7.136 | 1, 990.795 | 0.008 | 2.170 | 1.229 | 3.831 |
| | Fate = 2 (renewed) | 0 | | | N/A (reference category) | | | | |
| | Political ideology = 1 (liberal) | −2.211 | 0.525 | 17.724 | 1, 1415.253 | 0.000 | 0.110 | 0.039 | 0.307 |
| | Political ideology = 2 (centrist) | −1.956 | 0.388 | 25.357 | 1, 132.289 | 0.000 | 0.141 | 0.066 | 0.304 |
| | Political ideology = 3 (conservative) | 0 | | | N/A (reference category) | | | | |
| | Intercept | −0.707 | 0.229 | 9.565 | 1, 4967.547 | 0.002 | N/A | N/A | N/A |
| Medium priority | Fate = 1 | 0.291 | 0.200 | 2.118 | 1, 4116.470 | 0.146 | 1.338 | 0.904 | 1.979 |
| | Fate = 2 | 0 | | | N/A (reference category) | | | | |
| | Political ideology = 1 | −1.884 | 0.310 | 37.055 | 1, 926.314 | 0.000 | 0.152 | 0.083 | 0.279 |
| | Political ideology = 2 | −1.234 | 0.211 | 34.168 | 1, 1621.332 | 0.000 | 0.291 | 0.192 | 0.440 |
| | Political ideology = 3 | 0 | | | N/A (reference category) | | | | |
| | Intercept | −1.370 | 0.279 | 24.034 | 1, 2390.136 | 0.000 | N/A | N/A | N/A |

Overall model: $\chi^2$ = 91.253–106.685; df = 6; *p* = 0.000; AIC = 69.983–73.013; BIC = 105.317–108.081. Reference category is "high priority"; N/A = Not Applicable

| | **For the government** | | | | | | | | |
|---|---|---|---|---|---|---|---|---|---|
| | **Predictor** | **B** | **SE** | **F** | **df** | **Sig.** | **Exp (B)** | **Lower 95% CI** | **Upper 95% CI** |
| Low priority | Fate = 1 (destroyed) | 0.936 | 0.312 | 9.020 | 1, 530.926 | 0.003 | 2.549 | 1.383 | 4.697 |
| | Fate = 2 (renewed) | 0 | | | N/A (reference category) | | | | |
| | Political ideology = 1 (liberal) | −1.796 | 0.708 | 6.427 | 1, 273.470 | 0.012 | 0.166 | 0.041 | 0.667 |
| | Political ideology = 2 (centrist) | −1.700 | 0.411 | 17.143 | 1, 3149.353 | 0.000 | 0.183 | 0.082 | 0.409 |
| | Political ideology = 3 (conservative) | 0 | | | N/A (reference category) | | | | |
| | Religiosity | 0.394 | 0.140 | 7.904 | 1, 23.416 | 0.010 | 1.483 | 1.118 | 1.967 |
| | Intercept | −7.270 | 2.175 | 11.173 | 1, 23.457 | 0.003 | N/A | N/A | N/A |
| Medium priority | Fate = 1 | 0.269 | 0.218 | 1.527 | 1, 1261.227 | 0.217 | 1.309 | 0.854 | 2.005 |
| | Fate = 2 | 0 | | | N/A (reference category) | | | | |
| | Political ideology = 1 | −1.954 | 0.437 | 19.964 | 1, 1066.183 | 0.000 | 0.142 | 0.060 | 0.334 |
| | Political ideology = 2 | −0.862 | 0.241 | 12.796 | 1, 112.169 | 0.001 | 0.422 | 0.263 | 0.679 |
| | Political ideology = 3 | 0 | | | N/A (reference category) | | | | |
| | Religiosity | 0.103 | 0.062 | 2.755 | 1, 31.482 | 0.107 | 1.109 | 0.979 | 1.256 |
| | Intercept | −1.877 | 0.945 | 3.946 | 1, 32.317 | 0.056 | N/A | N/A | N/A |

Overall model: $\chi^2$ = 100.424–116.139; df = 8; *p* = 0.000; AIC = 208.627–277.668; BIC = 251.924–321.835. Reference category is "high priority"; N/A = Not Applicable

Nonetheless, it is important to realize that those who strongly agree the earth will be destroyed and also say climate change should be a low priority, whether for Christians or the government, are less than 3% of the full sample. Once again, political ideology is also significantly related to global warming attitudes, with political liberals and centrists being far less likely than conservatives to say that addressing global warming should be a low or medium priority for both Christians and the government to address (Table 6). Religiosity is also a significant predictor of attitudes about the government prioritizing global warming, with more religious respondents being almost one and a half times more likely to say that it should be a low versus high priority for each unit increase in the index (Table 6). The direction of relationships in this subsample are consistent with models using the full sample, and the effect sizes tend to be similar or slightly stronger.



## 4. Discussion

Our data suggest the relationships between Christian end time beliefs and attitudes about biodiversity and climate change are more nuanced than is typically understood or acknowledged in the broader environmental conservation and sustainability literature. The "end time apathy hypothesis" that suggests Christians do not care about the state of the environment because Christ will return and the earth will be destroyed is overly simplistic and biased to one particular eschatological perspective (Dispensationalism). It also does not adequately account for the pro-environmental views, initiatives, resources, and leadership that are increasingly widespread across the main branches of Christianity [16,17].

Most of our respondents reported the belief that Jesus Christ could return at any moment, but this belief was not significantly related to any of the biodiversity or climate variables we tested. This calls into question the efficacy of using the imminent return of Christ as a measure to distinguish between Christian end time beliefs. More compelling in this context is to compare those who think the earth will be destroyed and those who think the earth will be renewed. We found that close to twice as many respondents believe the earth will be renewed, not destroyed, at the end. This is notable as our sample is heavily focused on American evangelicals, who have previously been recognized for promoting dispensationalist end time theology predicting that the earth will be destroyed. At the same time, there is growing evidence that younger evangelicals in particular are aligning more with the belief that the earth will ultimately be renewed, not destroyed [32]. These younger evangelicals are also far more likely to hold pro-environment and pro-climate attitudes than their parents' generations [42,55].

In bivariate comparisons, beliefs about the fate of the earth were significantly associated with various biodiversity and climate change attitudes, consistent with other research (e.g., Pew Research Center [11]). Importantly, however, these relationships became non-significant when accounting for other religious and political variables in multivariate analyses. The final models consistently included biblical literalism and political ideology as key indicators. This generally fits with previous research on American Protestants (including evangelicals), which has identified theology and politics as the two primary reasons given for their negative climate and environmental attitudes [18].

Biblical literalism is often a stronger predictor of environmental attitudes than other religious variables including evangelical identification [2]. Biblical literalist views are often related to religious fundamentalism and conservative evangelicalism [37,56,57], and most studies find, as we do, that they are associated with less support for climate or biodiversity action [36,37,58]. This is often attributed not to views of the end times but rather to anthropocentric and dominionistic attitudes arising in part from more literal readings of the creation accounts in the Book of Genesis [10,18,59]. However, there are also multiple exceptions in the literature. Hempel and Smith [60] unexpectedly found that biblical literalism was positively associated with willingness to pay for environmental protection, and Chung et al. [56] found that it increased pro-environmental behavior even when controlling for other factors including political ideology. Further research is needed to understand when and why people's approach to interpreting the Bible influences different environmental measures.

While end time beliefs are generally insignificant in our study, there is an exception with those who hold stronger views about the fate of the earth. For this minority of respondents, their end time beliefs are significantly associated with how much they think addressing climate change should be a priority for both Christians and the government; those who strongly believe the earth will be destroyed think climate change should be less of a priority compared to those who think the earth will be renewed. While this relationship fits with the broader narrative around the "end time apathy hypothesis", it is important to recognize that those who strongly agree that the earth will be destroyed, while also saying global warming should be a low priority for Christians or the government, make up only about 3% of our sample. So, while this result is statistically significant, it is a different question as to whether it has real-world impacts. This relationship does not describe most Christians

in our study, or even most evangelicals. Regardless, it does suggest that future studies on end time beliefs should include measures of their strength or salience. This also fits with earlier findings that end time beliefs were most strongly associated with environmental attitudes among religious leaders/clergy compared with the public [37]. While eschatology could influence the attitudes of some Christians on these issues, end time beliefs can be complex, confusing, and distant: "End time views are a half-mile wide…but only an inch deep. Hardly anyone makes important day-to-day decisions based upon these end time views" (Jim Ball, former executive director of the Evangelical Environmental Network, cited by Veldman [12] (p. 40)). Many individuals may simply not understand or focus on end time beliefs enough to influence present-day concerns such as climate change and biodiversity loss.

An important finding is that the nuanced relationship between end time beliefs and climate change attitudes was not reflected in levels of concern about biodiversity, where it was consistently insignificant. This may be due to how politically polarized global warming has become relative to biodiversity loss. Additionally, climate solutions are often discussed at the international policy level, which may feed into a dispensationalist end time narrative about the rising up of a one-world government led by the Antichrist [38,39]. Although efforts to address biodiversity loss also involve extensive international negotiations, they have not risen to nearly the same levels of visibility and controversy as the United Nations Framework Convention on Climate Change and its highly publicized Conference of the Parties (COPs).

Furthermore, most studies have found that, even though religion may exert influence here, it is politics that plays the dominant role in shaping climate and environmental attitudes, including among American Christians [18,36,61]. Our findings add more evidence to support this understanding. We find political ideology to be more consistently significant and to have a stronger effect on our dependent variables than any of the other religious and sociodemographic variables in our study. In line with existing research, our results show that political conservatives are less worried and less supportive of prioritizing action on biodiversity and climate change than those who identify as liberal. At the same time, our findings that both religious and political variables are significant in the same model prompt questions about how these factors may synergize with each other, resulting in greater influence than they exert separately [18]. One manifestation of this potential synergistic relationship is Christian Nationalism, which is the merging of Christian and national identities. Christian Nationalism is particularly potent in the United States, where a growing body of research has found it to be a stronger and more consistent predictor of numerous socio-political views, including attitudes about science, than any other religious and political characteristic [62,63].

Another important avenue for future research is on the influence of end time beliefs among Christians outside the United States, particularly those in the majority world. Western Christian theology exerts considerable influence around the world for multiple reasons. Many of the centers of theological research, education, and publishing have been based in the West, which also has a robust track record of sending missionaries to spread the Christian faith in other countries. Further study is needed to understand how end time beliefs may shape environmental attitudes in contexts that have been influenced by Western theology but are further removed from American politics.

## 5. Conclusions

This study contributes to the sustainability literature by analyzing the relationship between major Christian end time beliefs and key environmental attitudes in the United States. We find that biodiversity and climate change attitudes among American Christians are highly associated with political ideology and biblical literalism, which is in line with much of the existing research. Furthermore, we find that end time beliefs are largely not related to levels of concern and support for action on biodiversity and climate change—at least among younger Christians—except among the minority with the strongest views

on the fate of the earth. This suggests that either there has been an overgeneralization of Christian views and their environmental implications or there is a shift underway between the views of Generation Z Christians and older generations (e.g., see the work of Lowe et al. [42] and the Pew Research Center [55]), or perhaps both. These relationships matter for socio-ecological sustainability due to the considerable influence Christians hold in American society and politics, and the implications this has for global conservation and sustainability efforts more broadly. The environmental community should take care to avoid overgeneralizing religious beliefs or uncritically perpetuating stereotypes of faith communities. This study supports the argument that contrary to how Christian end time beliefs have often been portrayed in popular American discourse, they may generally not be a major obstacle to caring about and supporting action on two of the greatest sustainability challenges of our time.

**Supplementary Materials:** The following supporting information can be downloaded at: https://www.mdpi.com/article/10.3390/su15119071/s1.

**Author Contributions:** Conceptualization, B.S.L., S.K.J., G.D.I. and A.L.P.; Methodology, B.S.L., S.K.J., G.D.I. and A.L.P.; Formal analysis, B.S.L. and G.D.I.; Investigation, B.S.L.; Data curation, B.S.L.; Writing—original draft, B.S.L. and S.K.J.; Writing—review & editing, B.S.L., S.K.J., G.D.I. and A.L.P.; Visualization, B.S.L.; Supervision, S.K.J.; Project administration, B.S.L.; Funding acquisition, B.S.L. and S.K.J. All authors have read and agreed to the published version of the manuscript.

**Funding:** This research was funded in part by the National Science Foundation (DGE-1842473) and the Religious Research Association (Constant H. Jacquet Research Award).

**Institutional Review Board Statement:** This study was conducted in accordance with the Declaration of Helsinki and approved by the University of Florida's Institutional Review Board for studies involving humans (Approved as Exempt on 12/6/2019 under IRB#201903125).

**Informed Consent Statement:** Informed consent was obtained from all subjects involved in the study.

**Data Availability Statement:** The data used in this article are available upon reasonable request from the corresponding author.

**Conflicts of Interest:** The authors declare no conflict of interest.

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
