# Peer review of "Association of Religious End Time Beliefs with Attitudes toward Climate Change and Biodiversity Loss"

_sustainability, doi:10.3390/su15119071_

Round 1

Reviewer 1 Report

Benjamin Lowe and colleagues provide a very interesting analysis of the relationship between the beliefs of Christian college students and their views on biodiversity loss and climate change. Their sample is 1,669 undergraduate students aged 18 to 23 from 35 church affiliated/sponsored Protestant Christian colleges in the United States. 1,520 of these respondents were Christians and included in their analysis. These individuals are disproportionately conservative, white, and female relative to the general US population. The authors administered a questionnaire that measured the students’ perceptions of the priority that global warming and biodiversity loss prevention and mitigation efforts ought to have in the respondent’s personal behavior and for the church community and civil government. They recorded many variables (e.g., whether the Bible is to be understood literally; political party affiliation), but most of the analysis focused on students that had beliefs in two mutually distinct views related the return of Jesus Christ: destruction of the Earth or renewal of the Earth.  In general, the authors found that their respondents tended to accept human caused global warming and worry about it to some extent, but an even higher percentage was worried about biodiversity loss. Further most accepted that Jesus Christ could return at any time, and many believed His return is eminent. 24% believed the Earth will be destroyed by fire when this happens, while 43% believed it will be renewed.

The authors compared end time beliefs to the priority the students believe should be given to biodiversity loss and global warming. End time beliefs are not statistically associated with different levels of prioritization of biodiversity loss, although beliefs regarding biblical literalism and (especially) political philosophy do impact the level of priority assigned to biodiversity. However, those who believe in the destruction of the Earth with Christ’s 2nd coming are more likely to say dealing with global warming should be a low priority for the Church community and at a governmental scale. Biblical literalism and (especially) political philosophy also impact the level of priority assigned to preventing/mitigating global warming. However, political affiliation/philosophy is a more significant factor. From this, the authors conclude that views of end times are not a particularly strong predictor of Christian views regarding the importance of biodiversity loss or global warming, contrary to links that have been suggested based on anecdote and assumptions.

The study is interesting and well-reasoned. The sample suffers from the issue of all such samples in that young adult college students may not be a representative sample of all US Christians, let alone all Christians everywhere. However, the authors do not claim that their sample is fully representative, and instead keep their conclusions modest, stating only that that their findings indicate that this sample does not reflect a link between end time perspectives and views towards biodiversity and global warming. The results are interesting and well supported, and I have no meaningful suggestions. As a result, I encourage you to publish this manuscript in its current form.

Author Response

Thank you very much for reviewing our paper in detail! We are encouraged by your positive assessment and recommendation that it be published in its current form!

Reviewer 2 Report

Eschatology is a "coat of my colors." Your reference is only one, rather extreme version, namely, apocalyptic eschatology, which is embraced by many contemporary fundamentalist Christians. For a typological delineation of eschatological doctrines, see R.H. Charles, Eschatology. The Doctrine of a Future Life in Israel, Judaism and Christianity. A Critical History (Schocken Books).

Author Response

Thank you very much for reviewing our paper! We are encouraged by your positive assessment and have added the helpful R.H. Charles reference to the revised manuscript.

Reviewer 3 Report

This is an impressive article grounded in nicely-focused research on students in US Christian colleges, leading to statistically well-evidenced conclusions challenging the view that beliefs regarding the end times have a significant impact on attitudes to biodiversity loss and climate change. Biblical literalism is shown to be a more significant factor. The paper is excellent on its own terms, although the authors recognize its limitations in terms of the age-group and geographical range of the respondents studied. I would encourage them to develop their research further through surveying older Christians and also those outside the US, but that would be a matter for future articles. There is also scope for exploring the wider historical development of end-times beliefs and their relationship to attitudes to the environment. 

Author Response

Thank you very much for reviewing our paper! We are encouraged by your positive assessment. Thanks in addition for your insightful suggestions on avenues for future research. We wholeheartedly agree that it would be instructive and worthwhile to expand down the road to examine older generations as well as populations outside the United States.

Reviewer 4 Report

The article submitted proposes an in-depth analysis of a very salient topic in the current sociological debate which analyze the contemporary relationship between religions and ecology. In particular, the paper examines a very interesting case-study which is the attitude of the Us evangelical young adults toward the climate crisis offering a new set of dates, and entering in dialogue with the larger literature published at this regard.

Precisely for the very specific nature of the case study presented, one of the critical aspects of the paper is revealed by the attempt of the authors to extend assumptions and evidences to the contemporary Christianity at large, instead of remaning on the specific case study.

In the abstract, at the beginning of the paper and especially in the conclusion persist an evident tendency to talk about Christian/Christianity as if it could be associated with the specific "climate-change attitude" revealed by the US-Evangelical.

For example, see line 421 -426. The biblical interpretation in favor of the environment is not a “surprisingly exception” for religious communities, there is a long tradition of Orthodox, Catholic, but also Protestant groups in Europe, which are involved in the Earth stewardship. In fact, looking at the Evangelical and Protestant context in Europe the Us-evangelical's attitude toward the environment seems to be the exception.

Moreover in the assertion at lines 459-464 the authors should pay more attention to all those Asiatic religious traditions which historically shaped societal environmental behaviors.

These are only two examples of how the paper should be rethought beyond the US-centric framework. Starting from the frequent abuse of the words Christian or Christianity.

Regarding this point I also invite the authors to explain why they especially considered the opinions regarding the “End of Time” as an indicator of sensibility/insensibility toward the climate change, considering that there could be many different biblical references to looking at.

Considering more technical aspects I suggest to the authors to give more information about the US school system, because not all the readers are familiar with the difference between college and university, or with which are the school programs of these typology of schools selected, having this information will help to get a clearer idea about the observed population.

I would also suggest indicating the research questions not in the discussion part, but in a separate paragraph (maybe with the methodology) to outline immediately and clearly which are the research objectives.

Please indicate in the footnote what Qualtrics is.

In conclusion the results are properly explained, but putting them into the right perspective (more contextualized with respect to the object of the study) will help to reinforce analytical and theoretical aspects.

Author Response

Thank you for your careful review of our paper and the detailed feedback you have provided. We have gone through very carefully to address your concerns and edit the manuscript accordingly.

Specifically, we have revised the language—especially in the abstract, introduction, and conclusion sections—to ensure we clearly focus on our sample population and avoid extending the results beyond this group.

This includes being explicit throughout that our study focuses on a particular subset (American Protestants) of the diverse world religion that is Christianity. We hope this addresses your concerns of "the abuse of the words Christian and Christianity" in our earlier version. Given that this study is focused on a particular branch of Christianity, we also do not address the relationship between sustainability issues and other religions, which are important but beyond the scope of this study.

Thank you for helpfully pointing out that readers may be confused about the differences between colleges and universities in the US context. We have revised this language to provide more clarification.

Finally, we agree with your suggestion to better identify the specific research questions at the beginning, and so we have now highlighted them at the end of the introduction.

Thank you again!

Round 2

Reviewer 4 Report

The authors  addressed all the suggested changes. The paper is accepted in the present form.